# Portability of a text mining algorithm for detecting adverse drug reactions in electronic health records across diverse patient groups in two Dutch hospitals

Britt W. M. van de Burgt [1,2]*, Loes F. C. van Dijck[3], Bjorn Dullemond[4],
Naomi T. Jessurun[5], Minou van Seyen[6], Rob J. van Marum[6,7,8], Remco J. A. van Wensen[9],
Wai-Yan Liu[9], Carolien M. J. van der Linden[10], Rene J. E. Grouls[1], R. Arthur Bouwman[2,13],
Erik H. M. Korsten[2], Toine C. G. Egberts[11,12]

1 Department of Clinical Pharmacy, Catharina Hospital Eindhoven, Eindhoven, The Netherlands,
2 Department of Electrical Engineering, Signal Processing Group, Technical University Eindhoven,
Eindhoven, The Netherlands, 3 Department of Clinical Pharmacy, Sint Jans Gasthuis, Weert, The
Netherlands, 4 Department of Mathematics and Computer Science, Technical University Eindhoven,
Eindhoven, The Netherlands, 5 Netherlands Pharmacovigilance Centre LAREB, 's-Hertogenbosch,
The Netherlands, 6 Department of Clinical Pharmacy, Jeroen Bosch Hospital, 's-Hertogenbosch,
The Netherlands, 7 Departments of Clinical Pharmacology and Geriatrics, Jeroen Bosch Hospital,
's-Hertogenbosch, The Netherlands, 8 Department of Elderly Care Medicine, Amsterdam Public Health
Research Institute, Amsterdam UMC Location VUmc, Amsterdam, The Netherlands, 9 Department
of Orthopaedic Surgery & Trauma, Catharina Hospital, Eindhoven, The Netherlands, 10 Department
of Geriatrics, Catharina Hospital Eindhoven, Eindhoven, The Netherlands, 11 Department of
Clinical Pharmacy, University Medical Centre Utrecht, Utrecht, The Netherlands, 12 Department
of Pharmacoepidemiology and Clinical Pharmacology, Faculty of Science, Utrecht Institute for
Pharmaceutical Sciences, Utrecht University, Utrecht, The Netherlands, 13 Department of Anesthesiology,
Catharina Hospital Eindhoven, Eindhoven, The Netherlands

* britt.vd.burgt@catharinaziekenhuis.nl

## Abstract

Adverse Drug Reactions (ADRs) pose a significant challenge in healthcare. While structured documentation of ADRs in electronic health records (EHRs) enables automated alerting, many ADRs are recorded as unstructured free-text, limiting detection. Text mining (TM) shows potential for extracting clinically relevant data from unstructured text. However, the portability of TM algorithms across different institutions and departments remains uncertain, due to variations in EHR structures and documentation practices. To enhance these general-purpose algorithms, evaluating their portability is essential for ensuring effective performance across diverse clinical settings. To evaluate the portability of a previously developed TM-based ADR identification algorithm by assessing its performance using EHRs from two different departments in two different hospitals. EHR free-text data from 62 hospitalized patients in the geriatric and orthopedic departments of two Dutch teaching hospitals were reviewed for ADRs via manual review and the TM algorithm. Performance was evaluated using F-score, sensitivity and positive predictive value (PPV), with comparisons across hospitals and departments. Manual review identified 359 unique ADRs. The TM algorithm

**Data availability statement:** All relevant data are within the manuscript and its Supporting Information files.

**Funding:** The author(s) received no specific funding for this work.

**Competing interests:** The authors have declared that no competing interests exist.

detected 534 potential ADRs (pADRs), 286 of which overlapped with manual review, yielding an F-score of 0.64, sensitivity of 80% and PPV of 54%. Performance was consistent across hospitals and departments. Notably, 26 pADRs identified by the algorithm were clinically relevant yet missed in manual review. This study demonstrates portability of the TM algorithm by identifying pADRs across different hospitals and departments without adaptations. These findings support its broader implementation potential for ADR detection in diverse healthcare settings.

## Author summary

Adverse Drug Reactions (ADRs) present a significant challenge in healthcare, with many being recorded as unstructured free-text in Electronic Health Records (EHRs). This study evaluates the portability of a text mining (TM) algorithm developed for ADR identification in EHRs, by assessing its performance across two departments in two Dutch hospitals. EHR free-text data from 62 patients in the geriatric and orthopedic departments were analyzed using manual review and the TM algorithm. The results showed that the TM algorithm demonstrated a good performance, with an F-score of 0.64, sensitivity of 80%, and positive predictive value (PPV) of 54%. Additionally, the algorithm identified clinically relevant ADRs that were missed in manual review. These findings suggest that the TM algorithm is portable across different clinical settings without requiring adaptation, highlighting its potential for broader implementation in ADR detection.

## Background and significance

Adverse Drug Reactions (ADRs) remain a significant concern in healthcare, contributing to patient harm and increasing healthcare costs. [1,2] It has been estimated that 3.6% to 6.5% of hospitalizations are associated with an ADR [3–6] and that 28% to 56% of all ADRs could have been prevented. [6–9] While prevention is often regarded as the primary approach to manage ADRs, addressing their recurrence is necessary to minimize patient harm. Inadequate re-prescription of medications can trigger repeated ADRs, highlighting the need for better monitoring and communication among healthcare providers. [10] To effectively prevent such recurrences, it is recommended to document ADRs as coded data in the patient's electronic health record (EHR). [11–13] This enables automatic alerts in the event of re-prescription a drug that previously caused the ADR. However, in the study of Wasylewicz et al. only 2% of the ADRs are documented as coded data in the designated module of the EHR. [14] This lack of structured documentation hinders the functionality of automated detection and alerting systems, increasing the risk that critical ADRs may be overlooked [11,13,15].

Text mining (TM), a natural language processing (NLP) approach, has shown potential in extracting clinically relevant information from unstructured text sources

such as nursing reports. [16–19] Furthermore, the integration of AI/ML with IoT security, highlights the critical role of advanced technologies in ensuring data protection and privacy in healthcare systems, including Electronic Health Records (EHR). [20] Despite its potential, the portability of TM algorithms across different settings has yielded mixed results. [21–28] Portability refers to the algorithm's ability to perform effectively in multiple clinical settings (e.g., various departments, hospitals) without significant adaptations. Variations in EHR structure, terminology, and documentation practices poses a significant challenge, necessitating careful evaluation before broader implementation [26,29–36].

In our previous work, we developed a TM algorithm using R-scripts combined with MedDRA and SNOMED-CT to identify potential ADRs (pADRs) in the free-text of a Dutch EHR. [14,37] While the algorithm was validated within a single hospital and limited to non-surgical departments, its performance in other departments and clinical settings has not yet been assessed, primarily due to potential variations in documentation policies, clinical practices, and patient demographics. [14,37] To enhance the algorithm's utility, it is essential to evaluate its portability.

This study aims to assess the portability of a TM-based ADR identification algorithm by applying it to patient data from a second Dutch hospital that utilizes the same EHR system. Additionally, we investigate the algorithm's performance across two distinct departments (surgical and non-surgical).

## Materials and methods

### Ethics statement

The study was declared not subject to the Research Involving Human Subjects Act (non-WMO) by the accredited Medical Research Ethics Committees United (approval number: AW22.056/W22.100) and approved by the local ethics review committees of both hospitals. Treating physicians identified eligible patients, who were then approached and provided with detailed study information. On inclusion days, all patients within the orthopedic department were informed about the study. Written informed consent was obtained from all participants.

### Study design, setting and population

The performance of the previously in CZE developed and described TM algorithm on non-surgical patients for identifying pADRs was assessed in two teaching hospitals (Catharina Hospital Eindhoven [CZE] and Jeroen Bosch Hospital 's-Hertogenbosch [JBZ], the Netherlands) using EHRs of two departments a surgical (orthopedics) and a non-surgical (geriatrics) in both hospitals and compared to manual review of the EHRs.

Inclusion criteria were patients hospitalized during November 2022 until December 2023 for a minimum of 24 hours in the geriatrics or orthopedics departments at either hospital. To ensure adequate power for statistical analysis, each department was required to include a minimum of 15 patients, resulting in a total of at least 60 participants across both hospitals. The sample size was determined based on a power calculation, utilizing data from two previously conducted studies. This calculation indicated that a cohort of this size would provide sufficient statistical power (0.9) to detect meaningful differences, with a Type I error rate of 0.05. Both hospitals utilize the HiX (ChipSoft B.V., Amsterdam, The Netherlands) EHR system. The CZE is using version 6.3, while JBZ is on version 6.2, there was no difference in text retrieval between the two.

For each included patient, all free-text data from their EHR, spanning the year prior to their most recent discharge, were extracted. Free-text data consisted of: consultations (which contains physician and paramedical reports, and outpatient visits or contact data like telephone consults), questionnaires (including nursing reports), microbiology reports, pathology reports, radiology reports and medical history. To assess the extent of ADRs captured in structured data, the ADR and complications modules were also reviewed. Scanned documents, such as referral letters (often PDF scans, rather than structured text), were excluded due to limitations in processing non-text-based information.

## Identification of ADRs

For all included patients, EHRs were searched for ADRs in two ways, 1) manual EHR review (reference method) and 2) with the TM algorithm (index test method). For both methods, the Dutch G-standard database was used to identify drugs, including all generic medicine names, trade names, group names registered in the Netherlands, as well as an in-house developed synonym medication list. All ADRs were classified using the Medical Dictionary for Regulatory Activities (MedDRA version 25.0) and Systematized Nomenclature of Medicine Clinical Terms (SNOMED-CT). Serious ADRs were categorized using the European Medicines Agency's Important Medical Events list and causality for al identified ADRs was assessed using the Naranjo algorithm. Only ADRs with a Naranjo score of ≥1 (possible, probable, or certain) were included.

**Manual EHR review.** Manual review was performed independently by two assessors (a clinical pharmacist in training (BB) and a pharmacist in training (LD) following the protocol outlined by Wasylewicz et al. [14] The EHRs were searched for free-text notes containing ADRs, defined by the World Health Organization (WHO) as "a response to a drug which is noxious and unintended, and which occurs at doses normally used in man for the prophylaxis, diagnosis, or therapy of disease, or for the modifications of physiological function." [38] Symptoms or diseases with multifactorial causes, including medication, were considered ADRs (e.g., "hyponatremia due to malnutrition and hydrochlorothiazide use"). Duplicate ADRs were removed, and text fragments involving multiple symptoms counted as two ADRs (e.g., *opiates causing nausea and constipation*). Text fragments involving two drugs were counted as one ADR (e.g., "*low blood pressure because of metoprolol/ hydrochlorothiazide"*). In cases where consensus between the two assessors was not reached, a third assessor (a clinical pharmacologist/ clinical pharmacist (RG)) made the final decision.

**Textmining algorithm.** As an index test method, the TM algorithm was used after manual review. The same EHR extracts were reviewed by the algorithm. The earlier developed algorithm was coded in R (Version 4.5.0, The R Foundation, Auckland, New-Zeeland [39]) and has been described in detail elsewhere. [14,37] In brief, the algorithm was initially coded in Gaston Pharma (Gaston Medical, Eindhoven, The Netherlands), a rule-based clinical decision support system, using 5 key strategies to identify EHR notes containing possible ADRS. This was tested on 45 patients in the study by Wasylewicz et al. [14] In the second study by Van de Burgt et al. [37], the algorithm was recoded and improved in R, using the same population to be able to identify and categorize pADRs. Hereafter MedDRA and SNOMED-CT were added to identify possible ADRs. The last step contained adding R-script to the algorithm to present unique ADRs.

## Data collection ADRs

The following data was collected for each ADR: symptoms/reaction, involved medication, trigger word, MedDRA term, surrounding paragraph/context, date and registering healthcare professional. If an ADR was registered in the EHR module, the status of the ADR being approved by a physician or a pharmacist and the severity of the ADR were also documented. The anonymized data were recorded, edited and saved using Research Manager (Cloud9, Deventer).

In addition, characteristics such as sex and age were collected from the patients' EHRs. Medical history and laboratory results were collected to calculate the Charlson Comorbidity Index (CCI). The following information was collected to characterize the data: the median number of hospitalizations (>24 hours), the number of ambulatory visits (<24 hours), the length of most recent hospital stay, number of involved medical specialties, the number of EHR notes and the number of words and characters of the EHR notes.

## Analysis

Portability was defined as the agreement between the two methods (manual review and the TM algorithm) across both hospitals (JBZ and CZE). The F1-score, as described by Hripcsak & Rothschild (2005) [40], with a threshold of 0.6, was used to assess this agreement, with portability considered acceptable if the F1-score is above this value, as demonstrated

in previous studies. [37,41] Statistical comparisons of the agreement between the manual review and the TM algorithm's results were conducted using an unpaired T-test on the F-score. All statistical analyses were performed using IBM SPSS Statistics, version 28 (IBM Corp., Armonk, NY).

Concordance between the two methods was defined as an ADR identified by both the manual review and the TM algorithm, even if there were slight differences in phrasing or context. For example: *hives on administration of penicillin* and *urticaria after penicillin* was considered a match. Discrepancy was defined when the TM algorithm identified an pADR that was not noted in the manual review, or vice versa.

In cases of discrepancy, the pADR identified by the TM algorithm was reviewed for technical accuracy and clinical relevance by the clinical pharmacist in training (BB). If the pADR was deemed technically correct and clinically relevant (e.g., documented by a healthcare professional or acknowledged by the patient), it was considered a missed ADR by manual review.

The performance of the TM algorithm of the two distinct departments, surgical and non-surgical departments was evaluated by comparing its results with those from the manual EHR review. Standard performance metrics were used for the performance of the algorithm, including F-score, sensitivity and positive predictive value (PPV). To test whether the non-surgical department demonstrated a significantly higher F-score than the surgical department, a one-tailed t-test was conducted, based on the hypothesis that non-surgical department, with its more complex patient population, would document more ADRs, leading to higher sensitivity and PPV. A two-tailed test was also performed for completeness, and results were reported accordingly.

## Results

### Patient and data characteristics

In total 62 patients were included, whereof 32 of the JBZ (17 for the orthopedic department, 15 for the geriatric department) and 30 of the CZE (15 for both the orthopedic and the geriatric department). In the orthopedic department, inclusion led to two additional patients at JBZ who expressed their willingness to participate. This resulted in 25,684 free-text EHR notes for review.

Table 1 represents the patient and data characteristics divided per hospital and per department. Between the two hospitals, no significant differences were observed in patient or data characteristics. Only, the length of hospitalization was significantly longer in CZE compared to JBZ (11 days, [95-CI: 7.5 to 17.5] vs. 4 days, [95-CI: 2.0 to 14.0]; p = 0.02).

**Table 1. Characteristics divided per hospital (CZE vs. JBZ) and per department (geriatric vs. orthopedic).**

| Variable | All patients (n = 62) | | CZE (n = 30) | | JBZ (n = 32) | | P-value | Geriatric (n = 30) | | Orthopedic (n = 32) | | P-value |
|---|---|---|---|---|---|---|---|---|---|---|---|---|
| Age in years, mean (range) | 76 | (33-94) | 74.6 | (33-94) | 77.3 | (48-92) | 0.65 | 85.6 | (71-94) | 66.9 | (33-84) | <0.001 |
| Female, % | 50 | | 56.7 | | 43.8 | | 0.31 | 56.7 | | 43.8 | | 0.31 |
| Charlson Comorbidity Index at last hospitalization, median (range) | 4.0 (0-9) | | 4.0 | (0-9) | 4.3 | (0-7) | 0.54 | 5.0 | (3-8) | 3.0 | (0-9) | <0.001 |
| Days of hospitalization, median (range) | 8.0 | (1-48) | 11 | (2-48) | 4 | (1-32) | 0.02 | 18 | (2-48) | 2.5 | (1-18) | <0.001 |
| Free-text EHR notes per patient, median (range) | 347 | (70-1,720) | 283 | (89-1,720) | 386 | (70-1,231) | 0.07 | 480 | (99-1,720) | 248 | (70-670) | <0.001 |
| Words per patient, median (range) | 12,058 | (1,921-104,157) | 12,632 | (1,921-104,157) | 11,472 | (2,650-59,949) | 0.80 | 21,626 | (4,783-104,157) | 6,861.5 | (1,921-39,923) | <0.001 |
| Characters per patient, median (range) | 82,248 | (14,759-574,699) | 86,070 | (14,759-574,699) | 72,264 | (15,360-341,328) | 0.47 | 138,705 | (28,324-574,699) | 45,393 | (14,759-280,235) | <0.001 |

Significant differences were observed between the two departments. Patients in the Geriatric department were older (85.6 years [95-CI: 83.3 to 87.8] vs. 66.9 years [95-CI: 62.7 to 71.1]; p<0.001), had higher CCI scores (5.0 [95-CI: 5.0 to 6.0] vs. 3.0 [95-CI: 2.0 to 3.5]; p<0.001), and experienced longer hospitalizations (18 days [95-CI: 14.0 to 23.5] vs. 2.5 days [95-CI: 2.0 to 5.5]; p<0.001) compared to the Orthopedic department. There was also significantly more EHR data in the Geriatric department, with more numbers of notes (480 [95-CI: 385.5 to 605.5] vs. 248 [95-CI: 196.0 to 313.9]; p<0.001), words (21,626 [95-CI: 3135.1 to 18,679.0] vs. 6,861.5 [95-CI: 5,617.0 to 8,773.0]; p<0.001), and characters (138,705 [95-CI: 18,416.8 to 105,910.5] vs. 45,393 [95-CI: 36,050.0 to 53,968.0]; p<0.001). This was found in both hospitals, see S1 Table.

### Identification of ADRs

**Inclusion of ADRs.** Fig 1 provides a flowchart showing the inclusion of ADRs discovered during the manual EHR review. A total of 747 ADRs were identified by the two assessors. During matching, 58 potential reADRs were detected and 234 ADRs were identified as duplicates. Excluding the duplicates and reADRs resulted in 455 unique ADRs. 61% (n=278) of the included ADRs was identified by both assessors, 18% (n=81) was identified by one of both assessors and included after consensus discussion between both assessors and/or third assessor. 96 of the ADRs identified by one of the assessors were not included. This resulted in 359 unique ARDs. No ADRs with a Naranjo score<1 were found, which resulted in 359 unique included ADRs remained. The assessors had a Cohen's Kappa of 0.58 of the included ADRs, indicating moderate agreement. The discrepancies in ADR identification included variances in how clinical terminology was interpreted and the subjective nature of certain notes, which were resolved through consensus or by a third assessor.

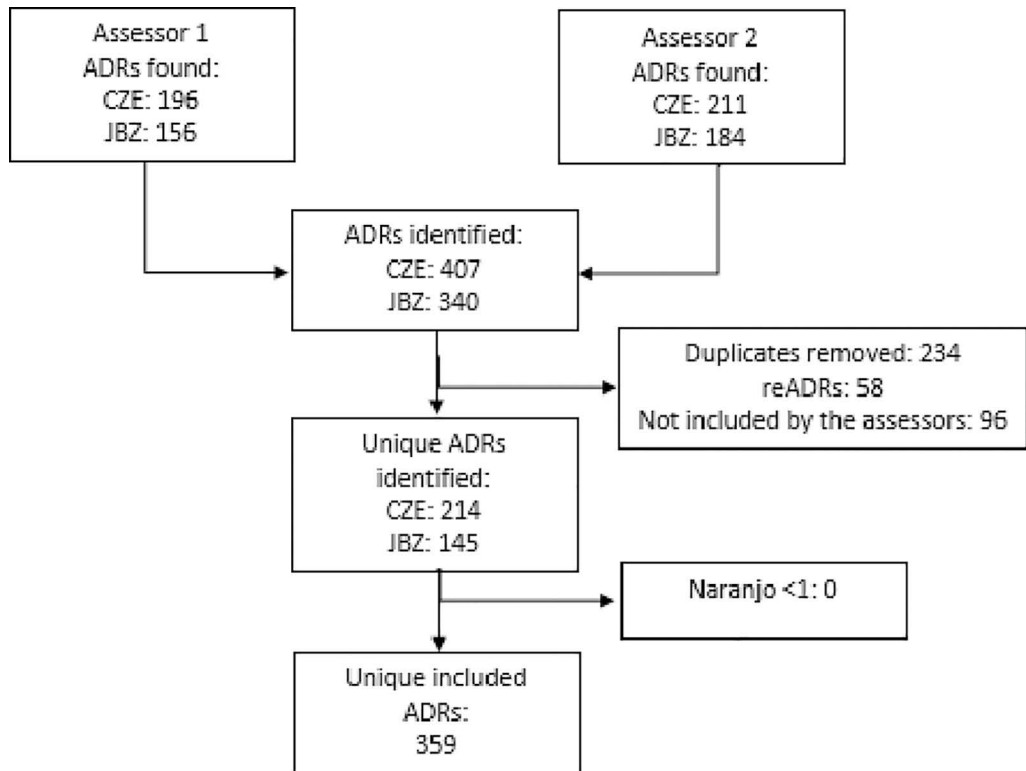

**Fig 1. Inclusion and exclusion of ADRs.**

**ADR Characteristics.** The median Naranjo score for the included ADRs was three (range 1–5). Most ADRs were judged possible (score 1–4), seven were scored probable (score 5–8) and null ADRs were scored definite (score ≥9). Overall, patients had a median of four unique ADRs (range 0–30). The median number of ADRs per patient in the CZE was six (range 0–30) and in the JBZ three (range 0–21). Approximately one-fifth (19.2%, n = 69) of all ADRs were classified as potentially serious. Overall, ADRs were also recorded in a structured EHR field in 21% of cases. More specifically, 50 ADRs (23%) were recorded in a structured format at CZE, compared to 26 ADRs (18%) at JBZ.

ADRs were found in different types of EHR notes. Most of the unique included ADRs (85%, n = 304) were cited in physician notes. However, 13% (n = 48) of the ADRs were found in nursing notes. The remaining identified ADRs were found in EHR notes of dieticians (n = 5) and physiotherapy notes (n = 2). ADRs included into the study were recorded by individual healthcare professionals, dived over 18 medical specialties.

**Comparison of manual EHR review and TM algorithm.** The algorithm identified a total of 534 potential pADRs, of which 286 were in concordance with those identified in the manual EHR review. The remaining 248 pADRs were identified by the algorithm but not found in the manual review, representing discrepancies. Notably, 26 of these 248 discrepancies had a Naranjo score ≥1, indicating that they were missed by the manual review. The median Naranjo score was thee (range 1–5), with most judged as possible, four were scored probable and null were scored definite. Additionally, three of the 26 discrepancies identified by the algorithm were classified as serious. These serious pADRs were retinopathy, hemorrhage and heart failure.

Of the 359 ADRs identified during the manual review, 286 were in concordance with the algorithm, resulting in a sensitivity of 80% and a PPV of 54%. Of the 69 potentially serious ADRs identified in the manual review, the algorithm correctly identified 53 (77%).

Fig 2 displays Venn diagrams illustrating the overlap between the EHR review methods for the total dataset, JBZ, CZE, and both orthopedic and geriatric departments. Table 2 provides the F-score, sensitivity and PPV for the total dataset as well as for each hospital and department.

## Portability of the algorithm in two different hospitals

The F-score, sensitivity, and PPV for both departments are presented in Table 2, while Fig 2 shows the Venn diagrams illustrating the overlap between the EHR review methods. For the CZE, the algorithm identified 313 pADRs, of which 169 were in concordance with those identified in the manual EHR review (which identified 214 ADRs), and 144 pADRs were classified as discrepancies. This resulted in an F-score of 0.64, a sensitivity of 79% and a PPV of 54%. Additionally, the algorithm identified 14 extra pADRs with a Naranjo score ≥1.

For the JBZ, the algorithm identified 221 pADRs, of which 117 were in concordance with those identified in the manual EHR review (which identified 145 ADRs), and 104 pADRs were classified as discrepancies. This resulted in an F-score of 0.64, a sensitivity of 81% and a PPV of 53%. Additionally, the algorithm identified 12 extra pADRs with a Naranjo score ≥1.

No significant difference in F-score between the hospitals were found (t = -1.423; p = 0.162; 95%-CI -17.62 to 3.04), despite the CZE (M = 70.2; SD = 19.1) having more ADRs than the JBZ (M = 62.9; SD = 15.2).

## Portability of the algorithm in two different departments (Orthopedic (surgical) vs Geriatric (non-surgical))

For both departments see Table 2 for the F-score, sensitivity and PPV and Fig 2 for the Venn diagrams illustrating the overlap between the EHR review methods. For the geriatric department, the algorithm identified 422 pADRs, of which 223 were in concordance with those identified in the manual EHR review (which identified 274 ADRs), and 199 pADRs were classified as discrepancies. This resulted in an F-score of 0.64, a sensitivity of 81% and a PPV of 53%. Additionally, the algorithm identified 17 extra pADRs with a Naranjo score ≥1.

For the orthopedic department, the algorithm identified 122 pADRs, of which 63 were in concordance with those identified in the manual EHR review (which identified 85 ADRs), and 49 pADRs were classified as discrepancies. This resulted

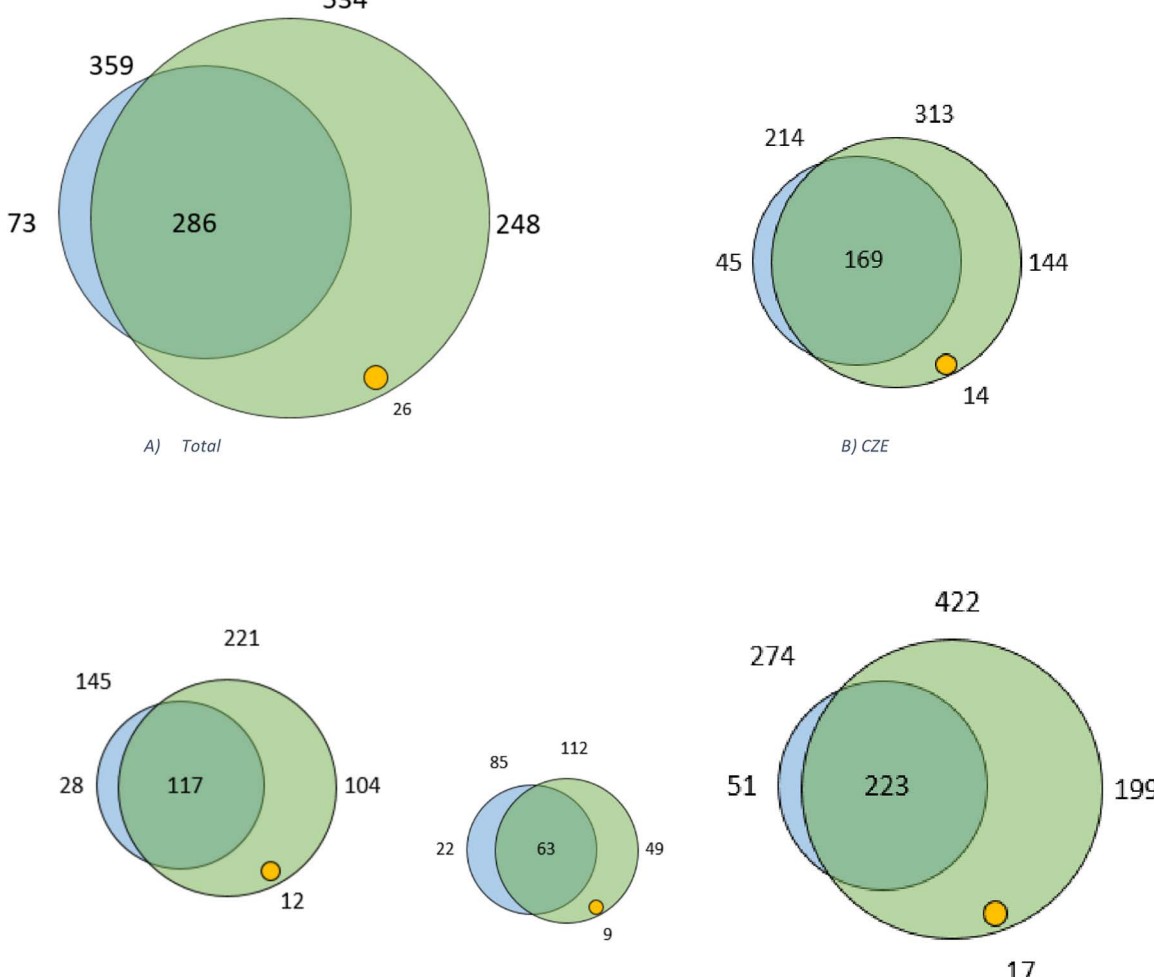

**Fig 2. Venn diagram presenting unique (possible) adverse drug reactions ((p)ADRs) of a) total, b) CZE, c) JBZ, d) Orthopedic department and e) Geriatric department.** The blue circle including the green portion, represents the total number of unique ADRs identified by the manual electronic health record (EHR) review. The green circle represents the total number of unique pADRs identified by the TM algorithm. The overlap between green and blue represents the number of concordance. The green represents the discrepancies, wherein the yellow circle represents pADRs found only by the TM algorithm. The blue alone represents missed ADRs by the algorithm.

**Table 2. TM algorithm versus manual review: The F-score, sensitivity and PPV of total, for both hospitals and departments.**

|                  | Total          | CZE            | JBZ            | Geriatric      | Orthopedic  |
|------------------|----------------|----------------|----------------|----------------|-------------|
| F-score          | 0.64           | 0.64           | 0.64           | 0.64           | 0.64        |
| Sensitivity (%)  | 286/359 (80)   | 169/214 (79)   | 117/145 (81)   | 223/274 (81)   | 63/85 (74)  |
| PPV (%)          | 286/534 (54)   | 169/313 (54)   | 117/221 (53)   | 223/422 (53)   | 63/112 (56) |

in an F-score of 0.64, a sensitivity of 74% and a PPV of 56%. Additionally, the algorithm identified 9 extra pADRs with a Naranjo score ≥1.

The difference in F-score between the orthopedic department (M = 72.3; SD = 19.2) and the geriatric department (M = 62.6; SD = 15.3) was not statistically significant (t = 1.87; p = 0.068). The 95%-confidence interval includes zero (-0.75 to 20.17), suggesting that the result was not statistically significant in a two-tailed test. However, based on the hypothesis that the Geriatrics department would document more ADRs due to the complexity of the patient population, a one-tailed p-value (0.034) suggests a potentially significant difference in the expected direction. See S1 Fig and S2 Table for the Venn-diagram, PPV, sensitivity and F-score per hospital per department.

## Discussion

This study provides valuable insights into the portability of a TM algorithm for identifying pADRs in EHRs. By assessing the algorithm's performance across two hospitals and two different departments, we explored its ability to function effectively in varied clinical settings with no adaptations. While our results demonstrate promising sensitivity and precision, variability in documentation practices and data characteristics between institutions highlighted challenges.

### Portability across hospitals

The TM algorithm performed similarly across the two hospitals, with no difference in F-scores. The comparable sensitivity (CZE: 79%, JBZ: 81%) and PPV (CZE: 54% vs. JBZ: 53%) suggest that the algorithm effectively identified pADRs in both settings, despite minor discrepancies in data characteristics such as length of hospitalization and number of EHR notes. These findings indicate that the algorithm is adaptable within institutions using the same EHR system.

One discrepancy in the data characteristics was the length of hospitalization, which was significantly longer in CZE compared to JBZ. This difference is likely attributable to variations in the clinical indications for admission and the times of admissions in the included year. At CZE, a specialized hospital for cardiac surgery and endocarditis, two patients experienced prolonged hospitalization due to endocarditis, a condition that typically requiring prolonged inpatient care. These two patients were not statistical outliers. Several other patients in CZE also had longer hospital stays due to multiple admissions. As a result, the overall length of stay in CZE was higher than in JBZ.

### Portability across departments

The algorithm's performance varied between the geriatric and orthopedic departments, with a higher sensitivity observed in the geriatric cohort (81% vs. 74%), though PPV remained similar (53% vs. 56%). The differences in documentation volume and complexity likely contributed to these findings, as the geriatric department had significantly more words and longer EHR notes and longer hospital stays. Although the overall F-score differences of the two-tailed t-test were not statistically significant, the one-tailed test may suggest a potentially meaningful variation, warranting further investigation in larger cohorts.

### Clinical practice

Our findings highlight the role of structured ADR documentation in facilitating automated detection and alerting mechanisms. Of the identified pADRs, 13,9% (n = 58) were reADRs, consistent with the findings of Zhang et al and Welk et al. who reported reADRs occurrences between 10–30%. [42,43] Furthermore, in our previous study of Wasylewicz et al., only 2% of ADRs were identified using the dedicated ADR module, meaning they were available in a structured format suitable for clinical decision support system (CDSS) alerts. Despite increased efforts within the JBZ and CZE, through continuing research to enhance ADR documentation using the ADR module, the proportion of ADRs registered in this structured manner across these hospitals only increased to 21%. This is comparable to the less than 15% rate of ADEs explicitly linked to specific drugs of the study from Kopacheva et al. [44] This underscores the opportunity for improvement: by

using algorithmic tools to support and optimize the documentation process, structured registration rates could potentially be increased to 80%. This may enhance the prevention of reADRs and mitigate patient harm.

Additionally, the algorithm demonstrated its additional value by, successfully detecting 26 pADRs missed during manual review. It's important to emphasize that no physician would typically perform such manual reviews due to time constraints and limited return on investment. Therefore, any contribution from the algorithm is a clear benefit, as long as false positives remain manageable. False positives where mainly due to three reasons: 1) Incorrect association of the medication with the ADR; 2) Replacement of one drug by another for medical reasons; 3) Patient refusal of medication due to potential side effects. The improvements we would suggest are: incorporating a step to exclude medications based on specific indications, adding a more nuanced rule for medication changes, ensuring the system recognizes these as legitimate clinical decisions and developing a filtering mechanism to differentiate between patient-reported concerns about medications and actual ADRs. Although the algorithm generated some false positives, the PPV remained above 50%, supporting its applicability in clinical practice. [37] This suggests that the algorithm can provide valuable assistance in detecting pADRs without overwhelming clinicians with too many irrelevant alerts. Further improvements in specificity could be achieved by enabling either humans (e.g., human-in-the-loop) or artificial intelligence to assess the context, thereby filtering out unlikely ADRs [45].

Another potential benefit of this study for clinical practice is the opportunity to perform benchmarking between different hospitals. By comparing practices and outcomes, institutions can leverage each other's efforts, share insights, and adopt best practices to enhance patient care. Moreover, applying the algorithm across multiple hospital settings enables the identification of previously unrecognized and rare ADRs, further contributing to improved medication safety.

## Limitations

Several limitations should be acknowledged. First, while the algorithm performed well in the studied hospitals, its generalizability to other EHR systems and languages remains uncertain. Future studies should explore its portability across diverse healthcare settings with different EHRs. Second, the study did not include scanned documents, thereby potentially missing pADRs, particularly referral letters from the general practitioner that often contain ADR information. Additionally, we did not consider potential biases in documentation styles among different healthcare professionals. Language and terminology may vary depending on the provider's specialty. However, the EHR history included notes from both ambulatory visits and hospitalizations across multiple medical specialties (n = 18).

Another limitation concerns causality assessment. Naranjo was used to assess causality of the pADRs. However, its reproducibility and validity are debated, as no universally accepted method exists for causality assessment of ADRs. [46] Moreover, only seven pADRs had a probable score, and null were scored definite. Despite these limitations, Naranjo was the preferred method in this study due to its wide acceptance and simplicity, making it a practical tool for causality assessment in clinical research.

## Future directions

It is important to acknowledge the broader issue of ADR underreporting by healthcare professionals. [47] While TM algorithms for detecting ADRs in free-text can enhance identification, they cannot fully address this challenge. Efforts should focus on improving ADR registration, wherein electronic reminders are in accordance with the "five rights" can improve the registration. [45,48,49] One potential solution involves integrating rule-based approaches with text mining. The study by Mahendran and McInnes explored three relation extraction techniques for ADR detection: rule-based methods, deep learning, and contextualized language models. [50] Their findings indicate that while contextualized language models achieved high overall performance, rule-based approaches outperformed them in certain relation types (such as frequency-Drug and ADR-Drug). Similarly, the review by van de Burgt et al. [45] highlighted the potential of combining rule-based methods with text mining to ensure the right information reaches the right person at the right time. This integration could enhance ADR registration through electronic reminders for healthcare professionals. Using ADRs recorded

in free text as input for CDSS could serve as an alert system for physicians, reducing unintended re-prescriptions and strengthening ADR reporting practices.

Our study complements existing research on automated ADE detection methods. A systematic review by Modi et al. [51] highlights various approaches for extracting ADEs from clinical notes, including rule-based, machine learning, and deep learning techniques. While these approaches have shown promise, challenges remain in balancing sensitivity and specificity, handling variations in clinical language, and integrating these methods into real-world healthcare workflows. Our findings indicate that our sensitivity of 80% and specificity of 54% are comparable to existing approaches, as demonstrated by consistent results across two hospitals. However, the feasibility of real-time implementation in clinical practice remains to be explored.

Artificial intelligence (AI) such as TM are being integrated with various technologies to transform healthcare. It is combined with EHRs, genomic profiles [52], and wearable health technologies to analyze complex datasets, predict disease progression, and recommend optimized treatment strategies. [53–57] For example, in medical imaging, the integration of AI into Picture Archiving and Communication Systems (PACS) has proven crucial, increasing diagnostic accuracy by up to 93.2% and reducing diagnostic times for critical conditions such as intracranial hemorrhages by up to 90%. [58].

Therefore, the final step in enhancing this algorithm is integrating it with a system that provides information in return to healthcare professionals, such as a CDSS. [45] The CDSS would complement the algorithm by retrieving structured ADR data, refining broad drug categories (e.g., "antibiotic") into specific medications (e.g., "penicillin"). However, it is crucial to ensure professional acceptance while implementing it real-time in clinical practice. This can be achieved by incorporating a "human in the loop". In practice, the algorithm will integrate with the CDSS, which, upon a patient's hospitalization, will trigger the analysis of patient notes to identify pADRs. The CDSS will then refine these pADRs, mapping general drug terms to specific medications. A healthcare professional will review and verify the potential ADR with the patient, and the confirmed ADR will be recorded in the patient's ADR table within the EHR. This integration enhances patient safety in the hospital by combining structured and unstructured EHR data to prevent (re)ADRs. It does however, leaves the challenge to ensure that the alerts generated by the TM-CDSS system can also reach healthcare providers across various settings, especially those using different EHR systems. The portability of alerts and their integration into diverse clinical environments remains a key hurdle to achieving comprehensive patient safety.

## Conclusion

This study demonstrates that the TM algorithm is portable and can identify pADRs across different hospitals and departments without adaptations. These findings highlight its broader implementation potential for ADR detection in diverse healthcare settings, especially when integrated with CDSS and clinician validation to enhance ADR registration, reduce medication risks, and improve patient safety.

## Supporting information

**S1 Table. All variables divided per hospital per department.**
(TIFF)

**S2 Table. PPV, sensitivity and F-score per hospital per department.**
(TIFF)

**S1 Fig. Venn-diagram concordance and discrepancies per hospital per department.**
(TIFF)

**S1 Data. Patient and EHR characteristics.**
(XLSX)

**S2 Data. Raw data pADRs, assessors, Naranjo, serious, EHR review and recorded as structured.**
(XLSX)

## Author contributions

**Conceptualization:** Britt van de Burgt, Remco J.A. van Wensen, Wai-Yan Liu, Carolien M.J. van der Linden, Rene J.E. Grouls, R Arthur Bouwman, Erik H.M. Korsten, Toine C.G. Egberts.

**Data curation:** Britt van de Burgt, Loes F.C. van Dijck, Naomi T. Jessurun.

**Formal analysis:** Britt van de Burgt.

**Investigation:** Britt van de Burgt, Rob J. van Marum.

**Methodology:** Britt van de Burgt, Naomi T. Jessurun, Minou van Seyen, Rob J. van Marum.

**Software:** Britt van de Burgt, Bjorn Dullemond.

**Supervision:** Rene J.E. Grouls, R Arthur Bouwman, Erik H.M. Korsten, Toine C.G. Egberts.

**Validation:** Britt van de Burgt.

**Visualization:** Britt van de Burgt, Rene J.E. Grouls.

**Writing – original draft:** Britt van de Burgt, Loes F.C. van Dijck.

**Writing – review & editing:** Britt van de Burgt, Minou van Seyen, Rob J. van Marum, Remco J.A. van Wensen, Wai-Yan Liu, Carolien M.J. van der Linden, Rene J.E. Grouls, R Arthur Bouwman, Erik H.M. Korsten, Toine C.G. Egberts.

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
