## [Decision Letter · Decision Letter 0]

17 Nov 2025

Response to Reviewers
Revised Manuscript with Track Changes
Manuscript
**Journal Requirements:**

1. Please ensure that your Ethics Statement is available in its entirety at the beginning of your Methods section, under a subheading 'Ethics Statement'. It must include:

1) The name(s) of the Institutional Review Board(s) or Ethics Committee(s)

2) The approval number(s), or a statement that approval was granted by the named board(s)

3) (for human participants/donors) - A statement that formal consent was obtained (must state whether verbal/written) OR the reason consent was not obtained (e.g. anonymity).

2. Please upload separate figure files in .tif or .eps format. Also, remove the figures from your manuscript file but keep the legends.

3. Please provide an Author Summary. This should appear in your manuscript between the Abstract (if applicable) and the Introduction, and should be 150–200 words long. The aim should be to make your findings accessible to a wide audience that includes both scientists and non-scientists. Sample summaries can be found on our website under Submission Guidelines:

https://journals.plos.org/digitalhealth/s/submission-guidelines#loc-parts-of-a-submission

4. We have noticed that you have uploaded Supporting Information files, but you have not included a list of legends. Please add a full list of legends for your Supporting Information files after the references list.

5. We notice that your supplementary figures and tables are included in the manuscript file. Please remove them and upload them with the file type 'Supporting Information'. Please ensure that each Supporting Information file has a legend listed in the manuscript after the references list.

6. We note that your Data Availability Statement is currently as follows: The data underlying this article are available in the article and in its online supplementary material.

7. Some material included in your submission may be copyrighted. According to PLOS’s copyright policy, authors who use figures or other material (e.g., graphics, clipart, maps) from another author or copyright holder must demonstrate or obtain permission to publish this material under the Creative Commons Attribution 4.0 International (CC BY 4.0) License used by PLOS journals. Please closely review the details of PLOS’s copyright requirements here: PLOS Licenses and Copyright. If you need to request permissions from a copyright holder, you may use PLOS's Copyright Content Permission form.

Potential Copyright Issues:

a. We do not publish any copyright or trademark symbols that usually accompany proprietary names, eg (R), (C), or TM (e.g. next to drug or reagent names). Therefore please remove all instances of trademark/copyright symbols throughout the text, including ‘Gaston Pharma®’ and ‘Research Manager®’.

**Additional Editor Comments (if provided):**
**Reviewers' Comments:**

**Comments to the Author**

1. Does this manuscript meet PLOS Digital Health’s publication criteria?

Reviewer #1: Yes

Reviewer #2: Yes

Reviewer #3: Yes

2. Has the statistical analysis been performed appropriately and rigorously?

Reviewer #1: Yes

Reviewer #2: Yes

Reviewer #3: N/A

3. Have the authors made all data underlying the findings in their manuscript fully available (please refer to the Data Availability Statement at the start of the manuscript PDF file)?

Reviewer #1: Yes

Reviewer #2: Yes

Reviewer #3: No

4. Is the manuscript presented in an intelligible fashion and written in standard English?

Reviewer #1: Yes

Reviewer #2: Yes

Reviewer #3: Yes

Reviewer #1: 1. Statistical comparison reporting of the differences in F-score could be clearer and more standardized. Specifically, reporting of the one-tailed $p$-value of $0.034$ suggesting potentially significant departmental difference after writing that the two-tailed test was not significant ($p=0.068$) adds a degree of ambiguity1111. This suggests the initial decision to use the two-tailed test might not cover the entire hypothesis of the authors or directional interpretation must be more convincingly justified.

2. Be systematic in the application of two-tailed tests, as the sample size is small and exploratory one-tailed tests are misleading in the absence of a well-defined a-priori hypothesis.

3. It is suggested for the authors to mention on the importance of AI/ML integration in healthcare/IoT security, which represents a subject of data management and security—a core issue with EHR systems.The recommended paper: "Personal Data Protection Model in IOMT-Blockchain on Secured Bit-Count Transmutation Data Encryption Approach". This paper in the Background subsection after Text mining (TM), a method of NLP, has been found to be useful in deriving clinically relevant information from unstructured text sources such as nursing reports.

4. Employ the same term pADR (potential Adverse Drug Reaction) for mentioning the algorithm output, and ADR for the concept or those ascertained by manual verification, especially in the titles of the Results section. For example, in Figure 2 legend, blue and green circles are referred to as "ADRs" where the study refers to them as "unique pADRs" in the text.

5. The authors cite the disputed validity of Naranjo's algorithm. Adding a brief statement that it was still used because it is one of the best accepted and simple-to-use methods would provide more explanation. Statistically correct the reportage in the Results and Discussion to express clearly the lack of significant difference in the F-score between departments based on the two-tailed T-test.

6. In mentioning Text Mining (TM)27, incorporate literature references to the cross-pollination of Machine Learning/AI with other technologies within healthcare. This will lay the ground for subsequent discussion on the usage of combining the TM algorithm with Clinical Decision Support Systems (CDSS).

Reviewer #2: The authors performed important work on portability/generalizability of general ADR retrieval from natural text.

Introduction:

First paragraph

The last two sentences of the first paragraph are overlapping it feels the authors are trying to convey the same thing twice., consider revision

The last sentence: HER instead of EHR

Second paragraph

The authors refer to mixed results of portability, it would be helpful to have some reference of the mixed results, and were these also on the topic of ADRs.

Third paragraph

The sentence of although sentence seems contradiction to itself. Please revise

Fourth paragraph

Why did the authors choose to validate in a second hospital with the same EHR system? Also a hospital within the same region?

No the question remains how the algorithm would perform in another Dutch region or different EHR structure.

Major: the measure/metrics of portability or the definition have not been defined in de the aim. When is it considered portable? What decline is considered acceptable? Please elaborate on this using references.

Methods

First paragraph

It is unclear from first read in which hospital the algorithm was devolved, also on what type of population.

And related why was this comparison chosen? Both general teaching hospitals.

Second paragraph

Consider moving Hospital and EHR specifics to first paragraph and merging the topics about inclusion from paragraph three.

Fourth paragraph

It is confusing to read that referral letters for example would not be text based, please elaborate.

Also, are referral letters not digitally send by general practitioner in the Netherlands?

Identification of pADRs

Please elaborate a bit more about the in-house developed algorithm, and, or place a clear reference here to further in the text.

Textmining algorithm

It's unclear or there's an unclear reference to how the algorithm works and what it does. The reference is only to how the algorithm was initially designed and earlier that is used G-standard and MedDRA datasets. Please elaborate on this.

Analysis

Major: if the aim is to assess portability of the algorithm between institutions why is it defined as the agreement between manual and TM algorithm? The sentence is very confusing to read. Isn’t the comparison of the performance between institutions and department types the real metric for portability?

Please elaborate who and did the concordance matching take place?

Results

First paragraph

Please elaborate on inclusion days, moreover consider moving this part to methods section

Second paragraph

Second sentence seems to end abruptly, please revise

The comparison between CZE and JBZ in length of stay ( median) is remarkable as the amount of notes and words is even higher in JBZ, please check if this is really correct. If this correct please elaborate on this in the discussion section also on the difference in notes.

Portability of the algorithm in two different hospitals

First sentence, please revise due to style

Discussion

Please elaborate on how the results ‘” highlight challenges related to variability in

documentation practices and data characteristics” as results seem very comparable.

Major: please elaborate on the 20% missed ADRs by the TM algorithm. What further steps need to be taken to improve sensitivity even further? This also comes back to how the algorithm works and which techniques can be equipped or mixed to improve outcome.

Reviewer #3: In this work, the authors present valuable research exploring the portability of a text-mining algorithm designed to detect potential adverse drug reactions (pADRs) in free-text clinical notes. Evaluations were undertaken in two hospitals and two clinical specialties, with the algorithm remaining unmodified throughout these assessments. The identification of ADRs was performed by manual review and the algorithm. The study revealed that performance was comparable across sites and departments. Furthermore, the tool successfully identified clinical ADRs that had been overlooked during manual review. The authors conclude that the algorithm is portable across institutions using the same EHR platform.

The results provide interesting findings related to the question if NLP algorithm for ADR detection can be used across hospitals. However, there are some opportunities for improvement:

- Even though testing across two hospitals makes the work more generalizable than single-site studies, only 62 patients were included. While the inclusion of over 25,000 free-text EHR notes provides a rich data source for analysis, the small number of patients per subgroup limits generalizability. The manuscript would improve by including an explanation why this sample size is sufficient or expanding the cohort.

- For manual review an agreement of 65% is reported but reasons for discrepancies are unclear. Providing a formal agreement measure and a brief explanation of disagreement sources would strengthen confidence in the manual review as the study’s reference standard.

- The results rely primarily on p-values without consistent reporting of confidence intervals. Including 95% CIs for key comparisons would enhance transparency and help readers assess the precision and practical significance of the findings.

- A PPV of 54% means half of the flagged pADRs were incorrect, but this is not analyzed. The inclusion of examples of common false positives and discussing ways to reduce them would be interesting.

**Do you want your identity to be public for this peer review?** For information about this choice, including consent withdrawal, please see our Privacy Policy

Reviewer #1: No

Reviewer #2: No

Reviewer #3: No

**Figure resubmission:**

**Reproducibility:** To enhance the reproducibility of your results, we recommend that authors of applicable studies deposit laboratory protocols in protocols.io, where a protocol can be assigned its own identifier (DOI) such that it can be cited independently in the future. Additionally, PLOS ONE offers an option to publish peer-reviewed clinical study protocols. Read more information on sharing protocols at https://plos.org/protocols?utm_medium=editorial-email&utm_source=authorletters&utm_campaign=protocols

---

## [Editor Report · Decision Letter 1]

24 Jan 2026

Portability of a text mining algorithm for detecting adverse drug reactions in electronic health records across diverse patient groups in two Dutch hospitals

PDIG-D-25-00649R1

Dear MSc van de Burgt,

We are pleased to inform you that your manuscript 'Portability of a text mining algorithm for detecting adverse drug reactions in electronic health records across diverse patient groups in two Dutch hospitals' has been provisionally accepted for publication in PLOS Digital Health.

Best regards,

Bushra Ali Sherazi

Guest Editor

PLOS Digital Health

**Additional Editor Comments (if provided):**

All comments were addressed to a satisfactory level. The manuscript is in publishable state.